# Description of the rates, trends and surgical burden associated with revision for prosthetic joint infection following primary and revision knee replacements in England and Wales: an analysis of the National Joint Registry for England, Wales, Northern Ireland and the Isle of Man

Erik Lenguerrand,[1] Michael R Whitehouse,[1] Andrew D Beswick,[1] Andrew D Toms,[2] Martyn L Porter,[3] Ashley W Blom,[1]  on behalf of the National Joint Registry for England, Wales, Northern Ireland and the Isle of Man

► Prepublication history and additional material are available. To view these files please visit the journal online (http://dx.doi.org/ 10.1136/bmjopen-2016-014056).

EL and MRW contributed equally.

For numbered affiliations see end of article.

**Correspondence to**
Professor Ashley W Blom;
Ashley.Blom@nbt.nhs.uk

## ABSTRACT

**Objectives**  To describe the prevalence rates of revision surgery for the treatment of prosthetic joint infection (PJI) for patients undergoing knee replacement, their time trends, the cumulative incidence function of revision for PJI and estimate the burden of PJI at health service level.

**Design**  We analysed revision knee replacements performed due to a diagnosis of PJI and the linked index procedures recorded in the National Joint Registry from 2003 to 2014 for England and Wales. The cohort analysed consisted of 679 010 index primary knee replacements, 33 920 index revision knee replacements and 8247 revision total knee replacements performed due to a diagnosis of PJI. The prevalence rates, their time trends investigated by time from index surgery to revision for PJI, cumulative incidence functions and the burden of PJI (total procedures) were calculated. Overall linear trends were investigated with log-linear regression.

**Results**  The incidence of revision total knee replacement due to PJI at 2 years was 3.2/1000 following primary and 14.4/1000 following revision knee replacement, respectively. The prevalence of revision due to PJI in the 3 months following primary knee replacement has risen by 2.5-fold (95% CI 1.2 to 5.3) from 2005 to 2013 and 7.5-fold (95% CI 1.0 to 56.1) following revision knee replacement. Over 1000 procedures per year are performed as a consequence of knee PJI, an increase of 2.8 from 2005 to 2013. Overall, 75% of revisions were two-stage with an increase in use of single-stage from 7.9% in 2005 to 18.8% in 2014.

**Conclusions**  Although the risk of revision due to PJI following knee replacement is low, it is rising, and coupled with the established and further predicted increased

### Strengths and limitations of this study

► This study provides updated and nationally representative descriptive evidence for England and Wales on revision for prosthetic joint infection (PJI) following knee replacements.
► Over 600 000 index primary replacements, 30 000 index revision replacements and 8000 revision total replacements performed due to a diagnosis of infection are included in the analyses.
► Contemporary evidence regarding changes over time in the risk of revision for PJI, its burden, and the types of revision surgery used for managing PJI is provided.
► The diagnosis of PJI recorded in the registry is not validated against a recognised international standard and may therefore be subject to over-reporting or under-reporting.
► Not all recorded revisions for PJI had an index primary or aseptic revision total knee replacement recorded in the National Joint Registry.

incidence of both primary and revision knee replacements, this represents an increasing and substantial treatment burden for orthopaedic service delivery in England and Wales. This has implications for future service design and the funding of individual and specialist centres.

## BACKGROUND

Deep prosthetic joint infection (PJI) is a rare but serious complication of total knee replacement. Patients experience severe pain, functional difficulties, poor quality

of life and, if untreated, loss of the affected limb or death.[1–3]

Estimates of the incidence of PJI after primary knee replacement have a range between 0.85% in Germany, 1.0% in the UK, 1.4% in Finland and 2.2% in the USA.[4–7] In a study from Finland, about 1.14% of patients had a PJI within 2 years of their primary knee replacement with a further 0.27% of patients presenting with PJI later than 2 years after primary surgery.[6] After aseptic revision and more complex knee replacement surgeries, infection rates may be considerably higher.[8] There is a paucity of evidence from UK centres and the available studies are old and based on single-centre data.[4 9]

Bacteria adhere as biofilm to implants and periprosthetic tissues making the treatment of PJI difficult.[10] Patients with PJI after knee replacement and their treating surgeons face complex and protracted treatment pathways. Initial treatment may involve surgical debridement, antibiotics and implant retention (DAIR), particularly in early postoperative infections. However, about 45%–52% of patients receiving DAIR may subsequently need revision of their implants.[11] Rates of revision of implants following treatment with a DAIR may be lower with strict selection criteria,[12] but larger single-centre cohort studies suggest the rate remains around 20% by 2 years.[13] Delays to effective infection control inherent in this process may lead to a poor postreplacement outcome.[14]

For patients with acute PJI diagnosed before biofilm maturation occurs, DAIR with modular exchange is a reasonable treatment option but for the majority of patients outside of this window of opportunity, contemporary management involves surgical revision with extensive debridement and antibiotic treatment with either immediate implant replacement (single-stage) or delayed implant replacement (two-stage).

The two-stage revision procedure was the first method reported for treatment of PJI of the knee that aimed to restore a functioning, painless knee.[15] The period between operations allows targeted antibiotic treatment and monitoring of clinical status and inflammatory markers. However, patients require two distinct planned major surgeries and a period of several months with limited knee function and protracted distress, concern and uncertainty.[16] An alternate one-stage strategy was developed by the Endo-Klinik and its use reported from 1976.[17] Key to the single-stage revision strategy is extensive debridement, identification of the infecting organisms and appropriate tailored antibiotic treatment.[18]

Pooled evidence from case series suggests similar rates of reinfection after each treatment.[19] The decision to treat with a single-stage or two-stage procedure may be guided by microbiological tests, patient and surgical factors but ultimately the choice of revision method is largely that of the treating surgeon.[2]

The National Joint Registry (NJR) for England, Wales Northern Ireland and the Isle of Man is the largest joint replacement database in the world and has collected data since April 2003.[20]

We analysed this dataset to assess the burden of revision surgery for knee PJI between April 2003 and December 2013. In this period, data were available from England and Wales. Our specific aims were to:

► Present the surgeon's perspective by describing the prevalence rates of revision surgery for the treatment of knee PJI following primary and revision surgery for aseptic indications, and their time trends broken down by time from index surgery to revision for infection.
► Present the patient's perspective by deriving cumulative incidence functions by type of index surgery.
► Estimate the burden of revision surgery for knee PJI at a health service level by accounting for all of the registered revisions and re-revision surgeries performed for knee PJI.

## METHODS
### Data source
In this observational study, we report analyses of data from the NJR.[20] The registry was established in 2003 and includes details of primary and revision knee replacements performed in England and Wales. Data entry for Northern Ireland and the Isle of Man did not commence until 2013 and 2015, respectively, and data linkage for those periods is limited, therefore they are excluded from this analysis.

### Index surgeries and revision surgeries for PJI
We grouped procedures as index knee replacements and revision surgeries for treatment of knee PJI. Index surgeries included all primary procedures and all revision procedures performed for an indication other than infection. The index revision procedures have been labelled 'aseptic revision' procedures to indicate they were not performed due to PJI. All index surgeries performed between 1 April 2003 and 31 December 2013 were included (to allow a minimum 12-month follow-up).

Information on microbiology results is not recorded in the NJR and infected index procedures were identified using subsequent revision performed for an indication of PJI. The diagnosis and treatment strategy for PJI is at the discretion of the surgeon according to local protocols and available information at the time of surgery. Revision surgeries performed as a consequence of PJI between 1 April 2003 and 31 December 2014 were considered. Revisions not performed for PJI but that were performed on a knee previously operated on due to PJI were not considered as index procedures, but were used alongside the revision and re-revision procedures for PJI to define the 'burden of PJI.' Non-surgical management of PJI and surgical procedures where no revision or modular exchange of implants is performed are not recorded in the NJR.

Revision procedures are reported in the NJR as a single-stage, a stage 1 of a two-stage revision, a stage 2 of a two-stage revision procedure, a conversion to arthrodesis or an amputation. DAIR procedures with modular exchange are recorded in the NJR dataset as single-stage revision

procedures. To identify which procedures recorded as single-stage revisions were DAIRs with modular exchange as opposed to complete single-stage revisions where implants fixed to bone are also revised, the component level data were considered for the index and revision procedures. Implant component labels are compulsory for all NJR records, the minimum dataset forms also contain fields for components removed but this information is discretionary and limited to the brand removed, therefore does not allow component tracing unless a linked index procedure exists. Procedures were therefore defined on the basis of the data provided for components implanted at revision surgery. Procedures recorded as single-stage revisions where only modular components were added (termed 'meniscal component' in the minimum dataset) were defined as DAIRs with modular exchange. Those where implants fixed to bone were implanted ('femoral component', 'tibial tray' +/− 'patella') were defined as single-stage revisions. Debridements where the surgeon either elects to not exchange modular components when they are present or where modular exchange is not possible (eg, with monoblock polyethylene tibial components) are not captured in the NJR.

### Missing procedures

There were 2880 two-stage revision procedures recorded where the index surgery was a primary knee replacement; in 792 (27%) of these, only a stage 2 of a two-stage procedure was recorded in the NJR. This was also observed for 154 (26%) of the 589 two-stage revision procedures following an index aseptic revision procedure. Patients with incompletely registered two-stage procedures did not differ from those with complete information in terms of gender, body mass index (BMI) and American Society of Anesthesiologists (ASA) physical status classification system grade (see online supplementary table 1). For index primary knee replacement, patients with incomplete information were on average 1 year older than those with complete information (p=0.05). For aseptic revision surgeries, no age difference was observed between the two groups.

For incomplete procedures, the date of the stage 1 of a two-stage revision procedure was estimated. First we derived the relative weight of time elapsed between the index and stage 1 of a two-stage procedure by year and type of index surgery using patients with complete information: $100 \times$ (length of time$_{\text{index surgery-1st stage}}$/length of time$_{\text{index surgery-2nd stage}}$). We applied these weights to the length of time observed between the index and stage 2 of a two-stage procedure for those with incomplete information to obtain an estimate for the time between the index surgery and stage 1 of the revision surgery.

### Statistical analyses

Analyses were performed with Stata SE V.13.1 (StataCorp).

#### Surgeon perspective

Prevalence rates of index surgeries performed between 2003 and 2013 and subsequently revised for infection between 2003 and 2014 were derived by year and type of index surgery. This provides a 'surgeon' perspective of revision for PJI by describing the proportion of knee replacements which required revision surgery for the management of infection. The prevalence rates were plotted by time from index surgery to revision for infection (≤3 months, 3–6 months, 6 months to 1 year, 1–2 years, 2–3 years, 3–4 years and 5–6 years). For two-stage surgeries, the date of stage 1 of a two-stage revision was selected to indicate the date of revision for PJI. Log-linear regression, using the year of the index knee replacement as a continuous independent factor, was used to investigate overall linear trends between 2005 and 2013. This period was selected as over 85% (proportion of procedure records submitted to the NJR compared with the levy returns for the number of implants sold) of knee replacements performed in 2005 and over 99% of those performed from 2007 onwards had been recorded in the NJR; prior to 2005, the data capture of the NJR was <75%.[21] When evidence of a time trend was identified (year of surgery, p≤0.05), the year of surgery was reconsidered as a categorical variable using 2005 as the reference period. Estimated relative risk and related 95% CI quantified the relative increase in prevalence rates between the period of interest and 2005.

#### Patient perspective

To move from a surgical perspective to a patient perspective, cumulative incidence functions[22–24] were derived by type of index surgery (primary or aseptic revision). These provide the probability of being revised as a consequence of PJI within a specific time period following the index surgery while accounting for the time patients were at risk of being revised for a PJI and the competing risks of death and revision for an aseptic indication.

#### Healthcare service perspective

The overall 'burden of revision surgery for knee PJI' was analysed using all revision procedures performed between 2003 and 2014 for an indication of PJI as well as any subsequent re-revision procedures for PJI or another indication.

Typically, knee replacement survivorship is 10 years or more following surgery,[20] re-revision procedures performed for another indication than PJI during the observation period are therefore considered as a consequence of the earlier management of PJI.

Revision procedures for an indication of PJI with no index procedure recorded in the NJR (n=3064, figure 1) and their re-revision procedures were also accounted. Contrary to the previous analyses in which index surgeries performed in 2014 and their revision procedures were not considered to allow for a minimum of 1 year postoperative follow-up, revisions for PJI relating to index knee replacements performed in 2014 (n=122) were included in this analysis.

The overall burden was reported by the year and type of revision surgery. Stage 1 of a two-stage revision and stage 2 of a two-stage revision were considered as one procedure to

Revision replacements performed as a consequence of prosthetic knee joint infection (PJI) recorded in the NJR between 04/2003 and 12/2014

A. Index primary and index "aseptic" revision replacements performed between 04/2003-12/2013 and subsequent revision replacements for PJI.

B. Revision replacements for PJI performed on patients with no previous NJR record. The first revision for PJI was performed between 04/2003-12/2013. These procedures are used to derive the burden of PJI†.

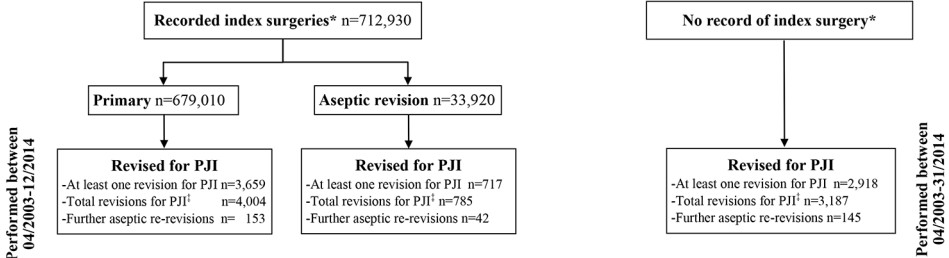

C. Additional index primary, index "aseptic" revision replacements performed between 01/2014-12/2014 and unlinked revision replacements for PJI performed between 01/2014-12/2014. These procedures are used to derive the burden of PJI† but not used in the other analyses as follow-up is < 1 year.

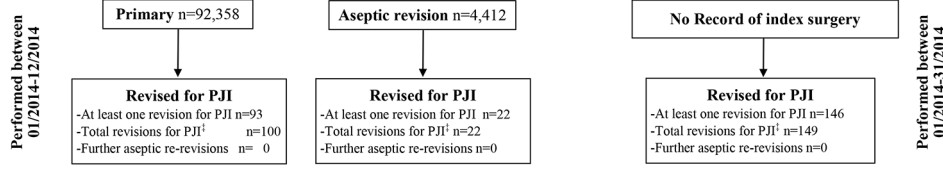

*An index surgery is defined as a joint replacement susceptible to be revised for PJI performed on patient with no previous history of revision for PJI.
‡This total includes the first recorded revision for PJI and any re-revision(s) for PJI. Stage one and stage two of the same two-stage revision are accounted as 1 procedure.
†The burden of PJI is defined as the procedures performed as a consequence of PJI: [4,004+785+3187+100+22+149]+[153+42+145]=8,587.

**Figure 1**  (A–C) Description of procedures recorded in the National Joint Registry (NJR).

avoid double counting. The same strategy was also used to account for multiple stage 1's of a two-stage revision procedure.

## RESULTS
### Surgeon perspective
There were 679 010 primary and 33 920 aseptic revision knee replacements recorded in the NJR for England and Wales between April 2003 and December 2013 (table 1).

A total of 3659 patients required at least one revision surgery due to PJI (4004 procedures) following a primary knee replacement (figure 1A). A further 717

patients required a subsequent revision for PJI (785 procedures) following an aseptic revision knee replacement (figure 1B). For 2918 patients who underwent revision for PJI (3187 procedures), no index surgery was recorded (figure 1C).

Of the primary knee replacements recorded in the NJR, 5.39/1000 (95% CI 5.21 to 5.56) were subsequently revised due to PJI (table 1). Of the aseptic revision knee replacements, 21.14/1000 (95% CI 19.61 to 22.67) were subsequently revised due to PJI.

Figure 2 shows trends in revision for PJI in the 2 years after primary knee replacement. Revision rates within 3 months

**Table 1**  Primary and aseptic revision knee replacement procedures revised for a knee prosthetic joint infection (PJI)

| Year of index procedure | Index primary replacement | | | Index aseptic revision replacement | | |
|---|---|---|---|---|---|---|
| | Total, N | Revised for PJI, n | Prevalence rate per 1000, 95% CI | Total, N | Revised for PJI, n | Prevalence rate per 1000, 95% CI |
| Total | 679 010 | 3659 | 5.39 (5.21 to 5.56) | 33 920 | 717 | 21.14 (19.61 to 22.67) |
| 2003 | 13 547 | 112 | 8.27 (6.74 to 9.79) | 534 | 24 | 44.94 (27.37 to 62.52) |
| 2004 | 27 706 | 184 | 6.64 (5.68 to 7.60) | 1001 | 24 | 23.98 (14.50 to 33.45) |
| 2005 | 41 848 | 314 | 7.50 (6.68 to 8.33) | 1612 | 41 | 25.43 (17.75 to 33.12) |
| 2006 | 49 482 | 346 | 6.99 (6.26 to 7.73) | 2094 | 55 | 26.27 (19.42 to 33.12) |
| 2007 | 66 618 | 471 | 7.07 (6.43 to 7.71) | 2847 | 60 | 21.07 (15.80 to 26.35) |
| 2008 | 73 859 | 443 | 6.00 (5.44 to 6.55) | 3538 | 87 | 24.59 (19.49 to 29.69) |
| 2009 | 75 612 | 465 | 6.15 (5.59 to 6.71) | 3884 | 91 | 23.43 (18.67 to 28.19) |
| 2010 | 78 276 | 423 | 5.40 (4.89 to 5.92) | 4196 | 102 | 24.31 (19.65 to 28.97) |
| 2011 | 81 795 | 378 | 4.62 (4.16 to 5.09) | 4336 | 81 | 18.68 (14.65 to 22.71) |
| 2012 | 85 161 | 341 | 4.00 (3.58 to 4.43) | 5115 | 93 | 18.18 (14.52 to 21.84) |
| 2013 | 85 106 | 182 | 2.14 (1.83 to 2.45) | 4763 | 59 | 12.39 (9.25 to 15.53) |

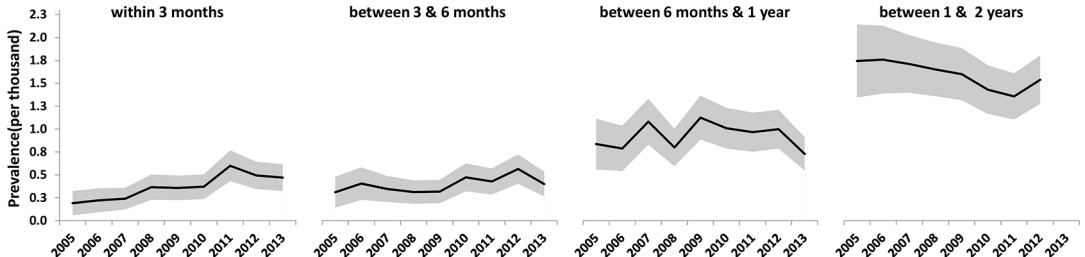

**Figure 2** Prevalence (95% CI) of revision for prosthetic joint infection within 2 years of the index primary knee replacement.

of the index procedure increased over time, with the prevalence rate in 2013 over twice that in 2005, rate ratio (RR) 2.46 (95% CI 1.15 to 5.25; time trend p<0.0001). No time trends for revision rates between 2005 and 2013 were apparent in the periods 3–6 months, 6 months to 1 year, and 1–2 years. Other than the prevalence rate of revision for PJI between 5 and 6 years which decreased over time, RR 0.53 (95% CI 0.33 to 0.86; time trend p=0.018), there was no evidence of time trends for the rates of late revision for PJI (see online supplementary figure 1).

Trends in subsequent revision of aseptic revision knee replacements for PJI are shown in figure 3 and online supplementary figure 2. The prevalence of aseptic revisions revised for PJI within 3 months of the procedure increased over time, RR 7.47 (95% CI 1.00 to 56.12; time trend p=0.001 for 2013 compared with 2006). No important differences were noted in the prevalence rates of revision for PJI performed at any time beyond 3 months from the index revision surgery (all time trends, p value>0.1) with the exception of revision for infection between 2 and 3 years for which the rates decreased over time, RR 0.40 (95% CI 0.18 to 0.88; time trend p=0.008 for 2011 compared with 2005).

### Patient perspective

Figure 4 shows the incidence over time of revision for PJI in patients with an index primary or aseptic revision while accounting for the risk of death and revision for any indication other than PJI. The probability of revision for PJI at 1 year following a primary knee replacement 1.7/1000 (95% CI 1.6 to 1.8) compared with 7.6/1000 (95% CI 6.8

to 8.6) following an aseptic revision knee replacement. At 2 years the probability was 3.2/1000 (95% CI 3.1 to 3.4) and 14.4/1000 (95% CI 13.1 to 15.7), respectively, and at 5 years it was 5.6/1000 (95% CI 5.4 to 5.8) and 24.1/1000 (95% CI 22.3 to 26.1), respectively. The probability of revision for infection within the first 10 years following primary knee replacement was 7.5/1000 (95% CI 7.2 to 7.8) and 31.3/1000 (95% CI 28.1 to 24.9) following an aseptic revision.

### Healthcare service perspective

Table 2 summarises the revision surgeries for the management of PJI and subsequent re-revisions (including repeated procedures to manage PJI and other revision procedures, see figure 1A–C) performed after primary and aseptic revision knee replacement by the year the (re-)revision was performed.

The absolute number of procedures performed as a consequence of PJI in England and Wales has increased from 378 in 2005 to 1048 in 2014, a relative increase of 2.8-fold. This is higher than the 2.1-fold increase in primary knee replacements over the same period, but similar to the 2.9 increase in aseptic revision surgeries between 2005 and 2013 (table 1).

Overall, 75% of revisions were conducted with a two-stage procedure but the use of single-stage revision for PJI has increased from 7.9% in 2005 to 18.8% in 2014. The median interval between stages in a two-stage revision following a primary index surgery was 99 days (25th–75th percentiles: 68, 156). A 101-day (67, 147) median interval was observed for two-stage revision performed to manage

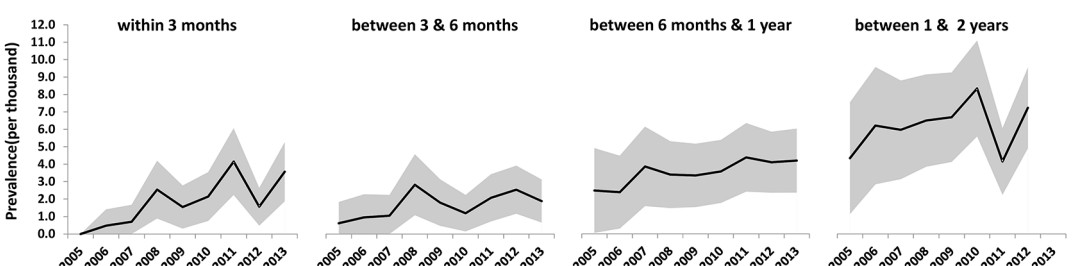

**Figure 3** Prevalence (95% CI) of revision for prosthetic joint infection within 2 years of the index aseptic revision knee replacement.

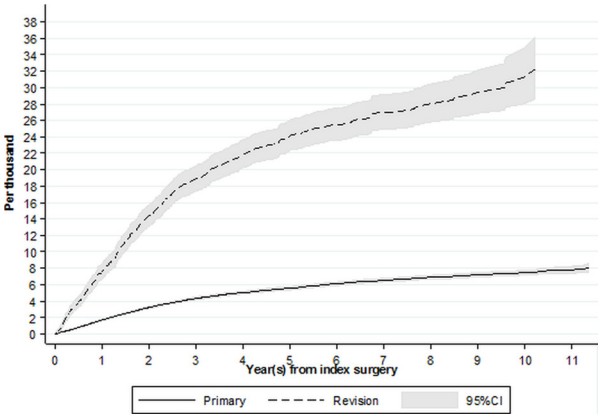

**Figure 4** Cumulative incidence function of revision for prosthetic joint infection following index primary and aseptic revision knee replacement.

PJI following an index aseptic revision surgery. The DAIR procedures represented around 5% of the total reported procedures and 9% of those reported in 2014. Other types of revision surgeries were rare and consisted mainly of arthrodesis (110 out of 134).

## DISCUSSION

This study is the largest to date investigating the treatment of PJI after knee replacement. It includes 712 930 primary and aseptic revision index procedures and 8247 revision total knee replacements performed due to a diagnosis of PJI. We have shown that the prevalence and incidence rates of revision for PJI are higher following aseptic revision knee replacement than for primary knee replacement and that the prevalence of revision for PJI

within 3 months of surgery has risen over time for primary and aseptic revision procedures. The total burden of treating PJI of the knee has risen substantially, with over 1000 procedures performed per year from 2011 onwards, which mirrors the greater rise in aseptic revision surgery compared with primary knee replacements.[20] From 2005, we have identified an increased use of single-stage as compared with two-stage revision for PJI of the knee which appears to have plateaued at approximately 20%.

### Strengths and weaknesses

The NJR is an established, large, prospective, observational arthroplasty register with comprehensive coverage of procedures undertaken, which is reassuring in terms of the generalisability of the data generated by the study. The data have some limitations. The recording of PJI as an indication for surgery is according to the opinion of the treating surgeon at the time of surgery and, as such, the diagnosis has not necessarily been referenced against a gold standard set of criteria and hence may be subject to misdiagnosis. While the NJR now represents a mature dataset with long-term follow-up, there are still a substantial number of procedures undergoing revision knee replacement where there is no record of a linked index primary or revision knee replacement or where there is an incomplete set of procedures recorded for two-stage revision surgery. This may be a reflection of the high survivorship of knee replacement for 10 years or more following surgery,[20] the fact that some PJI is acquired late and that not every case performed since 2003 in England and Wales has been recorded in the NJR. Given there was no difference between patients with complete two-stage episodes recorded and those with

**Table 2** Revision procedures performed as a consequence of PJI by type and year of procedure*

| Year of revision procedure | Total, N | DAIR, n (%) | Single-stage, n (%) | Two-stage, n (%) | Other†, n (%) |
|---|---|---|---|---|---|
| Total | 8587 | 408 (4.8) | 1589 (18.5) | 6456 (75.2) | 134 (1.6) |
| 2003 | 126 | 3 (2.4) | 116 (92.1) | 7 (05.6) | 0 (0.0) |
| 2004 | 236 | 3 (1.3) | 59 (25.0) | 171 (72.5) | 3 (1.3) |
| 2005 | 378 | 8 (2.1) | 30 (07.9) | 336 (88.9) | 4 (1.1) |
| 2006 | 485 | 11 (2.3) | 49 (10.1) | 419 (86.4) | 6 (1.2) |
| 2007 | 663 | 19 (2.9) | 70 (10.6) | 560 (84.5) | 14 (2.1) |
| 2008 | 783 | 29 (3.7) | 124 (15.8) | 615 (78.5) | 15 (1.9) |
| 2009 | 852 | 32 (3.8) | 145 (17.0) | 662 (77.7) | 13 (1.5) |
| 2010 | 931 | 38 (4.1) | 178 (19.1) | 707 (75.9) | 8 (0.9) |
| 2011 | 1004 | 51 (5.1) | 187 (18.6) | 748 (74.5) | 18 (1.8) |
| 2012 | 1035 | 58 (5.6) | 200 (19.3) | 754 (72.9) | 23 (2.2) |
| 2013 | 1046 | 61 (5.8) | 234 (22.4) | 732 (70.0) | 19 (1.8) |
| 2014 | 1048 | 95 (9.1) | 197 (18.8) | 745 (71.1) | 11 (1.1) |

*This table reports the revision procedures performed after any index surgery revised as a consequence of PJI including subsequent re-revision procedures whether performed to manage an infection or not. It also includes the revision procedures performed between 2003 and 2014 on 3064 patients with PJI but with no index procedure documented in the NJR.
†Conversion to arthrodesis or amputation.
DAIR, debridement, antibiotics and implant retention; NJR, National Joint Registry; PJI, prosthetic joint infection.

partial episodes recorded in terms of age, gender, BMI and ASA grade, we have assumed that the time interval of the former is generalisable to the latter, which may not be the case. The NJR does not capture information on microbiology results, non-surgical treatment of PJI and revision for PJI with implant retention but no modular exchange. Currently in the NJR, the only option to record DAIR procedures where modular exchange is performed is to record them as single-stage revision surgeries.[25] This complicates their differentiation from single-stage surgeries where the femoral and tibial components are revised. This represents a potential weakness of the data collection form as this may be subject to different interpretation by surgeons, despite the fact that recording of procedures in which any component is removed or inserted is mandatory. We have used component level data for individual cases to identify DAIR procedures with modular exchange. A few single-stage revisions for PJI unlinked to an index procedure were considered as DAIRs in view of the components implanted (n=66). The capture of revision surgeries, in particular those performed for PJI, is also not perfect in arthroplasty registries[26–29] and it is unclear how complete the capture of these procedures in the NJR is.

The annual burden of PJI with over 1000 procedures in recent years, while already high and expensive, is a conservative estimation and the cost of PJI considerable for the National Health Service.

## Comparison with other studies

Revision for infection is a rare complication of knee replacement. Our findings in this respect are consistent with previous studies from England and Wales[4 9] although the reported prevalence rates were higher in previous studies (around 1% at 5 years following primary replacement compared with 0.5% in our study). Our study was not contemporaneous with those studies (index surgeries performed between 1987 and 2001[9] or 1993 and 1996)[4] and the considered sample size in them was smaller (n=4788 and n=931). Evidence from other countries shows similar prevalence rates of revision for infection (rates ranging from 0.4% to 2.2%).[5–7 30 31] Studies that report incidence rates typically include any type of operations for the management of infection (not just those where implants are removed or changed) and this may be why they show higher rates (0.9%–1.4% between 1 and 5 years postoperatively).[6 32]

We have demonstrated a substantial time trend of an increased risk of revision for PJI within the first 3 months of an index primary or aseptic revision knee replacement being performed. This phenomenon is likely to be multifactorial. Factors that could lead to an increased diagnosis of PJI in this period and hence increased risk of revision include the increased accuracy of tests available to clinicians for the diagnosis of PJI,[33] coupled with more rapid diagnosis and/or treatment in specialist centres, the increased risk factors for PJI among the population undergoing knee replacement population (such as

elevated body mass index)[34] and increased bed occupancy within the healthcare setting in which these procedures were performed.[35] There may be a trend towards the use of revision surgery to manage PJI rather than suppressive treatment with antibiotics but we cannot comment on this as non-surgical management of PJI is beyond the scope of this study. There have been similar findings in other countries that could not be accounted for by risk factors recorded in those registries, suggesting this trend could reflect an actual general increase in the risk of PJI.[36] It is interesting to note from our results that there has not been a relative increase in the use of DAIRs with modular exchanges in the NJR to explain this phenomenon.

We have shown a greater incidence of revision for PJI following aseptic revision knee replacement (3.1% at 10 years) compared with primary knee replacement (0.8% at 10 years) consistent with the findings of previous cohort studies.[4 31 32 37–39] Previous surgery is a known risk factor for PJI,[40] this may be because of further bacterial contamination or increased length of surgery. Revision knee replacement tends to be performed in a population with increased host[41] and procedure risk factors for infection or further revision[42] and involve the introduction of a greater volume of prosthetic material and additional adjuncts such as bone graft that may present a favourable environment for bacterial colonisation and subsequent PJI.[43]

## Implications for clinicians and policymakers

The current rate of surgical revision of PJI of the knee stands at approximately 1000 cases per year across the NJR. This represents a significant healthcare burden which has more than doubled over the last decade, exceeding the rate of increase in primary knee replacements. The cost of revision knee replacement due to PJI is more than three times that of primary knee replacement and over twice that of aseptic revision.[44] In the National Health Service, the costs associated with revision knee replacement due to PJI have been shown to be in excess of £30 000 per case, greater than other indications for revision,[45] even before accounting for costs associated with litigation. Single-stage revision offers an advantage over the two-stage approach in terms of both patient-derived and surgeon-derived utility values in both the short term and long term in the hip[46] and may be associated with superior functional outcomes in the knee,[47] suggesting this increased burden could be ameliorated by the increased use of a single-stage revision strategy.

## Unanswered questions and future research

The observed trend of an increase in the risk of infection in the first 3 months following primary and aseptic revision total knee replacement is not currently well understood. Identification of the risk factors associated with this phenomenon and modifiable risk factors for PJI may help reduce this risk. Although the use of single-stage revision knee replacement for the treatment of PJI

has increased to approximately 20%, given the current evidence base shows equivalent reinfection rates following single-stage compared with two-stage revision surgery for PJI,[19 48] it might be possible that the proportion of patients treated with single-stage surgery taking account of patient, surgeon and operative factors could be further increased. Given the increasing burden of treating PJI, this may help ameliorate the increase in resources that will otherwise be required to treat this condition.[49] In the absence of data from randomised controlled trials or other direct comparisons, we plan further analyses of the NJR data which will compare outcomes after one-stage and two-stage methods with robust adjustment for key patient and surgical factors. Such evidence will support the decision-making process in the planning and treatment of patients with PJI after knee replacement.

**Author affiliations**
[1]Musculoskeletal Research Unit, University of Bristol, School of Clinical Sciences, Bristol, UK
[2]Exeter Knee Reconstruction Unit, Royal Devon and Exeter NHS Foundation Trust, Princess Elizabeth Orthopaedic Centre, Exeter, UK
[3]Centre for Hip Surgery, Wrightington Hospital, Wrightington, Wigan and Leigh NHS Foundation Trust, Lancashire, UK

**Acknowledgements** We thank the patients and staff of all the hospitals who have contributed data to the National Joint Registry. We are grateful to the Healthcare Quality Improvement Partnership, the National Joint Registry Steering Committee, and staff at the National Joint Registry for facilitating this work.

**Contributors** EL, MRW, ADB and AWB designed the study. The data were extracted by Northgate (Hemel Hempstead, UK). EL, ADB, MRW and AWB reviewed the published work. EL conducted the statistical analyses. All authors interpreted data and wrote the report. EL had full access to all the data and AWB is the guarantor.

**Funding** This article presents independent research funded by the National Institute for Health Research (NIHR) under its Programme Grants for Applied Research program (RP-PG-1210-12005). The funders had no role in study design, data collection and analysis, decision to publish, or preparation of the manuscript.

**Disclaimer** The views expressed represent those of the authors and do not necessarily reflect those of the National Joint Registry Steering Committee or Healthcare Quality Improvement Partnership, who do not vouch for how the information is presented. The views expressed in this article are those of the authors and not necessarily those of the NHS, the NIHR, or the Department of Health.

**Competing interests** MLP is the Medical Director of the National Joint Registry and also acts as Chair of the Programme Steering Committee for the National Institute for Health Research (NIHR) INFORM program (PGfAR program: RP-PG-1210-12005). The other authors have no conflicts of interest.

**Ethics approval** Patient consent was obtained for data collection by the National Joint Registry. According to the specifications of the NHS Health Research Authority, separate informed consent and ethical approval were not required for the present study.

**Provenance and peer review** Not commissioned; externally peer reviewed.

**Data sharing statement** Access to the data analysed in this study required permission from the National Joint Registry for England, Wales and Northern Ireland Research Sub-committee. http://www.njrcentre.org.uk/njrcentre/Research/Researchrequests/tabid/305/Default.aspx contains information on research data access request to the National Joint Registry.

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
