## [Reviewer comments · BMJ Open]

ARTICLE DETAILS

TITLE (PROVISIONAL)	Description of the rates, trends and surgical burden associated with revision for prosthetic joint infection following primary and revision knee replacement in England and Wales. An analysis of the National Joint Registry for England, Wales, Northern Ireland and the Isle of Man.
AUTHORS	Lenguerrand, Erik; Whitehouse, Michael; Beswick, Andrew; Toms, Andrew; Porter, Martyn; Blom, Ashley

VERSION 1 - REVIEW

REVIEWER	Professor Hemant Pandit University of Leeds, UK
REVIEW RETURNED	19-Sep-2016

GENERAL COMMENTS	The paper is relevant to a general reader and of course to an Orthopaedic surgeon. This work is timely and deserves publication. I have only some very minor comments to make... 1. Page 6 line 46: Please read the sentence would read better if developed is replaced by was developed.2. Page 8 line 24: is the date 1st January 2003 correct? I think there is a typo.3. The authors do not mention any information about the risk of death after PJI - they have access to the data and it will be a useful addition - clearly this is a burden to the clinicians, patients as well as health care service perspective.
---

REVIEWER	Werner Zimmerli Interdisciplinary Unit for Orthopaedic Infectious, Kantonsspital Baselland, University of Basel, Liestal, Switzerland.
REVIEW RETURNED	27-Sep-2016

GENERAL COMMENTS	Bmjopen-2016-014056 This is a study on the prevalence of periprosthetic joint infection (PJI) from the National Joint Registry for England, Wales, Northern Ireland and the Isle of Man with data from patients undergoing total knee arthroplasty (TKA) during a 12-year period from 2003-2014. General comments. As in most registry studies, the main advantage of the study is the large number of patients included in the analysis. This advantage is paid with a lack of precision, which cannot be reached without analyzing individual data of all patients included. Thus, the risk for underestimating the prevalence of PJI in this, as well as in other
---

registry studies, is considerable. Unfortunately, a number of important questions is not addressed in the present study, e.g.: (i) Which were the selection criteria for the choice of the surgical option (single stage vs two-stage? (ii) How many patients got another surgical option such as debridement with implant retention (DAIR) or arthrodesis? (iii) How many patients didn't get any surgical treatment, but got exclusive suppressive antimicrobial therapy for suppression? (iv) How long was the interval between explantation and implantation in patients with two-stage revision? (v) What was the spectrum of microorganisms? (vi) How many patients had polymicrobial PJI? (vii) How many patients had culture-negative PJI? If data for these questions were prospectively gathered in the Registry, they should be analyzed. The strengths of the study are (i) the observation of possible changes in the prevalence over time, and (ii) the observation of the infection rate during a long time period after surgery allowing to register also hematogenous PJI.

Specific comments.

1. Background/p.5: lines 26-29. The statement that "45-52% of patients receiving DAIR may subsequently need revision of their implants" does not reflect modern surgical management. If the patients for DAIR are strictly chosen according to an algorithm (NEJM 351:1645-54,2004, Clin Infect Dis 56:e1-25,2013), the failure rate is much lower (e.g. 5% in Clin Microbiol Infect 12:433-9,2006).

2. Background/p.5: lines 33-37. The statement "For the majority of patients with PJI...contemporary management involves ...immediate implant replacement...or delayed implant replacement". In different studies from specialized centers, a considerable fraction of patients with PJI after TKA was treated with DAIR (e.g. Clin Infect Dis 42:471-8,2006; Clin Microbiol Infect 12:433-9,2006: 52% DAIR). If PJI is actively looked for during the early postoperative period, many patients qualify for treatment with DAIR.

3. Background/p.5: lines 54-57. A randomized trial comparing the outcome of single-stage and two-stage revision would not be reasonable, because risk factors for failure with 1-stage treatment are well known. Thus, patients should be carefully selected for 1-stage exchange.

4. Methods/p.7: line38-40. As mentioned in the limitations, there is no definition of PJI in this study (see p.15).

5. Methods/p.7: line 40. The definition of "revision procedures" is not complete, since debridement with/without exchange of the modular parts of the device is also a revision procedure. The number of patients undergoing this procedure should also be given. In addition, the number of patients with PJI not undergoing any surgery at all, but only palliative antibiotic suppression should be given, otherwise the prevalence of PJI is underestimated.

6. Results/p.12: lines 3-5. The observation "Revision rates within 3 months of index procedure increased over time..." is very interesting. In the Discussion, possible reasons for this apparent increase should be analyzed (more rapid diagnosis? Better contact with rehabilitation centers leading to more rapid readmission? Less suppressive antibiotics without revision?).

7. Discussion/p.14: line 54. The statement "...utilisation of single stage revision could be safely increased" is somewhat dangerous, since the type of procedure should be carefully chosen according to an algorithm based on risk factors for the specific procedure. E.g. patients with sinus tracts or difficult-to-treat microorganisms should not be treated with 1-stage exchange.

8. Discussion/p.15: lines 37-46. The statement that "changing only modular components is not distinct from single stage surgery..." is not correct. In case of single stage exchange, the surgeon tries to

	eliminate the whole biofilm, which cannot be done when only modular parts are exchanged. 9. Discussion/p.17: lines 24ff. As mentioned above (#6), the discussion of the “increase...in the first 3 months...” is not complete at all.
--	---

REVIEWER	Ove Furnes Department of orthopaedic surgery Haukeland University Hospital Norway Head the Norwegian Arthroplasty Register
REVIEW RETURNED	20-Dec-2016

GENERAL COMMENTS	The paper Revision for prosthetic joint infection following primary and revision knee replacement and its overall surgical burden is an analysis of the National joint registry for England, Wales , Northern Ireland and the Isle of Man. It is an important study from the largest joint registry in the world. I have some suggestions and questions to the authors. The paper is difficult to read because of the complexity of the issue with prosthetic joint infections (PJI) and the terminology used. 1. Figure 1 The term: recorded index surgery is non-intuitive. To me index implies a primary operation but her it means both primary and subsequent revisions as long as they are not done due to infection. This should be explained. In the last row : performed between 01/2004-31/2014 the numbers of primary and aseptic revisions should be given. 2. In the abstract under the design heading 8031 revision total knee replacements are used, in table 2 the number of 8658 are used, in figure 1 I count 8302 PJI adding total revision for PJI. Explain this discrepancy. 3. The use of the term prevalence and cumulative revision risk is confusing for me. The authors use them differently in the results in the abstract, tables, figures and in the text in the results section. In the result section of the abstract the revision risk was 0.5% following primary and 2.1% following revision knee replacement. Do the authors mean prevalence or cumulative revision risk? Is the revision risk calculated after 1, 5 or 10 years following primary and revisions procedure? In the result section the authors refer to Table 1 for the revision rate but in this table the prevalence rate are given (5.39 per 1000). Is this the same as the revision risk of 0.5% in the abstract? I had expected that you used numbers from figure 4 as risk for revision. I would like to read the tables and figures and see the same numbers in the text. This is a general issue with the paper. 4. Another issue is the lack of distinction between DAIR (surgical debridement, antibiotic and implant retention), one stage and two stage revision for PJI. DAIR is discussed on page 15. It should be mentioned more clearly in the methods section. Some papers find up to 70% success with DAIR. Patients with acute PJI would usually go through a debridement with exchange of easily exchangeable parts like tibia polyethylene inserts. How do you handle an AGC
---

monoblock knee if it is revised due to infection with DAIR. Will this procedure be included in the register compared to a NexGen modular fixed bearing knee. How do you distinguish this procedure from a one stage revision? Can NJR distinguish between removal of implant, insert of implant, exchange of implant or implant parts, insert of cement spacer, exchange of polyethylene insert? This should be explained in the methods section (See Engesæter LB. Surgical procedures in the treatment of 784 infected THAs reported to the Norwegian Arthroplasty Register. Acta Orthop 2011;82 (5):530-537.

5. Datasource Paragraph 2: 1 st april or 1.january??
Index procedures should be explained including both primary and aseptic revisions. The explanation come after the term is introduced.

6. Statistical analyses. It is stated that 85% and 99% of knee replacements were captured. Indicate if these are primary cases or all, and give an estimate for the capture percentage of revisions.

7. Under patient perspective the authors state that competing risk of death and revision for an aseptic indication was accounted for. In the Fine and Grey model one can usually account for only one competing risk cause at the time. How did you manage this?

8. Results:
Figure 2 show increasing prevalence of PJI over time but table 1 show a decrease in prevalence by time. This discrepancy needs explanation?

9. Patient perspective
Probability of revision at 5 years was 0.56% and 2.41% which is not the same numbers as in the abstract. Explain.

10. Table 2 does not attempt to distinguish between one stage and DAIR where component parts are exchanged.

11. Discussion:
In the discussion under limitations and strengths is should be mentioned if NJR have validated the revision procedures and in particular PJIs
(See Gundtoft PH et al. Validation of the diagnosis prosthetic joint infection in the Danish Hip Arthroplasty Register. BJJ;98-B:320-5.

Gundtoft PH et al. One-year incidence of prosthetic joint infection in total hip arthroplasty: a cohort study with linkage of the Danish Hip Arthroplasty Register and Danish Microbiology Database. Osteoarthritis and Cartilage 2016 epub.

Lindgren JV et al. Validation of reoperations due to infection in the Swedish Hip Arthroplasty Register. BMC Musculoskeletal disorders 2014 Nov 19;15:384)

According to the NJR reports the yearly revision rate has increased since 2003. I have interpreted this as an increased completeness of reporting to NJR for revision operations. Could this be commented.

12. Can the increased use of modular implants be a cause of the increased revisionrate recorded for PJI in NJR?

REVIEWER	Gregory J. Stoddard University of Utah School of Medicine, USA
REVIEW RETURNED	26-Jan-2017

GENERAL COMMENTS	The authors have done a nice job of demonstrating the revision issue is a 0 to 3 months post-surgery phenomenon. In their Table 1, the prevalence of revisions is decreasing across time, suggesting, at first, that we are doing surgeries better as time goes on. In Figures 2 and 3, we see a relationship between revisions and temporal time in four panels, where the panels represent the time between surgery and revision. It is noticeable that the longer the time between surgery and revision, the flatter the trend line across years. So, revision risk is a 0 to 3 months post-surgery issue, and we are not getting better at reducing that risk. The prevalence of revisions across time in Table 1 is going down, but that is because >3 months data are being added to the denominator faster than the 0 to 3 month data. The authors recognized all of this, and the way they reported their conclusions correctly reflects it. I have no revision suggestions. Nice work!
---

VERSION 1 – AUTHOR RESPONSE

Reviewer: 1

Reviewer Name: Professor Hemant Pandit

Institution and Country: University of Leeds, UK

The paper is relevant to a general reader and of course to an Orthopaedic surgeon. This work is timely and deserves publication. I have only some very minor comments to make. We would like to thank Professor Pandit for his positive feedback. Each comment is addressed below.

1. Page 6 line 46: Please read the sentence would read better if developed is replaced by was developed.

RESPONSE:

The sentence has been modified as suggested.

2. Page 8 line 24: is the date 1st January 2003 correct? I think there is a typo.

RESPONSE:

The date has now been corrected to 1st April 2013.

3. The authors do not mention any information about the risk of death after PJI - they have access to the data and it will be a useful addition - clearly this is a burden to the clinicians, patients as well as health care service perspective.

RESPONSE:

We thank Professor Pandit for this suggestion. Unfortunately, presentation of mortality data would require linkage of the NJR data presented to the Office for National Statistics Data in order to establish date of death. We have not performed that linkage at this time and therefore are unable to add it to this manuscript. We feel it would be very difficult to fit the addition of the necessary work around the complex area of mortality into this paper given the current volume of information contained within the manuscript, and therefore feel that mortality is beyond the scope of this paper. The point is very pertinent though and this will be part of our ongoing future work in this area.

Reviewer: 2

Reviewer Name: Werner Zimmerli

Institution and Country: Interdisciplinary Unit for Orthopaedic Infectious, Kantonsspital Baselland, University of Basel, Liestal, Switzerland.

We would like to thank Professor Zimmerli for his thorough review of our manuscript, comments and suggestions.

General comments.

As in most registry studies, the main advantage of the study is the large number of patients included in the analysis. This advantage is paid with a lack of precision, which cannot be reached without analyzing individual data of all patients included. Thus, the risk for underestimating the prevalence of PJI in this, as well as in other registry studies, is considerable. Unfortunately, a number of important questions is not addressed in the present study, e.g.:

- (i) Which were the selection criteria for the choice of the surgical option (single stage vs two-stage?
- (ii) How many patients got another surgical option such as debridement with implant retention (DAIR) or arthrodesis? (iii) How many patients didn't get any surgical treatment, but got exclusive suppressive antimicrobial therapy for suppression?
- (iv) How long was the interval between explantation and implantation in patients with two-stage revision?
- (v) What was the spectrum of microorganisms?
- (vi) How many patients had polymicrobial PJI?
- (vii) How many patients had culture-negative PJI? If data for these questions were prospectively gathered in the Registry, they should be analyzed.

The strengths of the study are:

- the observation of possible changes in the prevalence over time, and
- the observation of the infection rate during a long time period after surgery allowing to register also hematogenous PJI.

RESPONSE:

As correctly stressed, analyses based on registry data are limited to the level of details associated with the data collectable at large scale, but this disadvantage is offset by the volume of available patients and the possibility to conduct temporal investigation. Case series have the exact opposite pros and cons, but mainly are lacking statistical power and more importantly representativeness. The NJR, one of the largest arthroplasty registry in the world, is no exception. It focusses on orthopaedic procedures during which a prosthesis or part(s) of a prosthesis are implanted, modified or replaced. Data on microbiology results, non-surgical management of PJI, surgery where no implants or accessories are added, taken away or modified, or on the selection criteria for single vs. two-stage surgery are not collected in the NJR and therefore cannot be analysed. As stated in the manuscript title, this research focusses therefore on surgical intervention for the management of PJI, i.e. revision surgery requiring the replacement of the implant or part of the implant, it does not claim to report the prevalence of PJI per se.

In the current classification used by the NJR, any change of part or all of an implant is classified as a revision. DAIR procedures that include a modular exchange are therefore recorded by the treating surgeon as a single stage revision on the data collection forms for upload of data to the NJR. We have now attempted to disentangle the DAIR procedures from the single-stage procedure using additional component level information of recorded cases. However, DAIRs where no modular exchange is performed (e.g. total knee replacements with monoblock polyethylene tibial components) are beyond the scope of the NJR and are not captured if no other implants or accessories (including bone cement) are added. Arthrodesis and amputation are reported in the "other" column of table 2. Despite those limitations, this work provides findings for very large numbers of procedures in England and Wales, countries for which the currently available published information on revision for PJI are

sparse, not representative and out-of-date.

To specifically address the above comments point by point:

i.

RESPONSE: The selection criteria for the choice of single stage vs. two-stage are at the discretion of the treating surgeon for that patient and procedure depending on local procedure and preference. The specific reason a surgeon may have chosen one treatment strategy over another is not recorded in the NJR. However, the two-stage approach remains the preferred treatment strategy in England and Wales as shown in this paper.

The following sentence has been added to the methods section:

“The diagnosis and treatment strategy for PJI is at the discretion of the surgeon according to local protocols and available information at the time of surgery.”

ii.

RESPONSE: Data on DAIR procedures with implant retention where no modular exchange is performed are not captured by the NJR as there has been no removal, addition or change of an implant or accessory. DAIR procedures where a modular exchange is performed are captured and recorded within the NJR as single stage procedures. We have now identified those procedures using component level data for individual cases, separated them from the “true” single stage revision procedures where femoral and tibial components fixed to bone are revised and amended table 2 to describe them. Arthrodesis procedures are described in the column “other” of table 2. During the observation period, only 110 of such procedures were used to manage PJI: 28 were unlinked to an index surgery, 20 were the first recorded revision for PJI following an index primary and 13 were recorded as the first revision for PJI following an index aseptic revision. The remaining 49 were re-revision procedures for PJI. Twenty-four amputations were finally added to the “other” procedures. The following sentences have been added to the results section (Healthcare service perspective paragraph):

“The DAIR procedures represented around 5% of the total reported procedures and 9% of those reported in 2014. Other types of revision surgeries were rare and consisted mainly of arthrodesis (110 out of 134).”

iii., v., vi., vii.

RESPONSE: Information on microbiology, non-surgical treatment of PJI or surgical treatment not involving the removal, addition or change of an implant are not captured by the NJR and are hence beyond the scope of this analysis.

This is now clearly acknowledged in the method section (Index surgeries and revision surgeries for PJI paragraph):

“Information on microbiology results are not recorded in the NJR...”

Non-surgical management of PJI and surgical procedures where no revision or modular exchange of implants is performed are not recorded in the NJR.”

iv.

RESPONSE: The following sentence has been added to the results section (Healthcare service perspective paragraph):

“The median interval between stages in a two-stage revision following a primary index surgery was 99 days (25th-75th percentiles: 68, 156). A 101 day (67, 147) median interval was observed for two-stage revision performed to manage PJI following an index aseptic revision surgery.”

Specific comments.

1. Background/p.5: lines 26-29. The statement that “45-52% of patients receiving DAIR may subsequently need revision of their implants” does not reflect modern surgical management. If the patients for DAIR are strictly chosen according to an algorithm (NEJM 351:1645-54,2004, Clin Infect Dis 56:e1-25,2013), the failure rate is much lower (e.g. 5% in Clin Microbiol Infect 12:433-9,2006).

RESPONSE:

Thank you for this suggestion. We have now modified the background section as follow:

“However about 45-52% of patients receiving DAIR may subsequently need revision of their implants.[11] Rates of revision of implants following treatment with a DAIR may be lower with strict selection criteria [12], but larger single centre cohort studies suggest the rate remains around 20% by two years [13].Delays to effective infection control inherent in this process may lead to a poor post replacement outcome. [14]”

2. Background/p.5: lines 33-37. The statement “For the majority of patients with PJI...contemporary management involves ...immediate implant replacement...or delayed implant replacement”. In different studies from specialized centers, a considerable fraction of patients with PJI after TKA was treated with DAIR (e.g. Clin Infect Dis 42:471-8,2006; Clin Microbiol Infect 12:433-9,2006: 52% DAIR). If PJI is actively looked for during the early postoperative period, many patients qualify for treatment with DAIR.

RESPONSE:

Thank you for this comment. We have now modified this sentence as follow:

“For patients with PJI after knee replacement who are diagnosed before biofilm formation occurs, DAIR with modular exchange is a reasonable treatment option but for the majority of patients outside of this window of opportunity, contemporary management involves surgical revision with extensive debridement and antibiotic treatment with either immediate implant replacement (single-stage) or delayed implant replacement (two-stage)”

3. Background/p.5: lines 54-57. A randomized trial comparing the outcome of single-stage and two-stage revision would not be reasonable, because risk factors for failure with 1-stage treatment are well known. Thus, patients should be carefully selected for 1-stage exchange.

RESPONSE:

While we agree that patients characteristics should be central to the selection-process between single and two-stage approaches, findings from the latest meta-analysis have shown no significant difference in the risk of re-infection between those two types of procedures at two years follow up in unselected patients (Kunutsor et al., PloS One 2016;11(3): e0151537). The principle of equipoise between single and two-stage revision is therefore supported by the most recent well conducted meta-analysis, supporting the need for such a trial. Our group is already conducting a multicentre, and now multinational, randomised controlled trial of single vs. two-stage revision for PJI of the hip (Strange S, Whitehouse MR, Beswick AD, Board T, Burston A, Burston B, et al. One-stage or two-stage revision surgery for prosthetic hip joint infection – the INFORM trial: a study protocol for a randomised controlled trial. Trials 2016;17(1):90.) This trial has met with broad acceptance and support from our public and patient involvement group, research departments, ethics committees, surgeons, hospitals and national funding bodies. We are in the process of developing a similar trial for the knee, which has received similar levels of support.

The findings of older systematic reviews are conflicting, two are in favour of the single stage approach (Masters et al., BMC Musculoskelet Disord. 2013;14:222, Nagra et al., 2016; 24(10):3106-3114) and the other is in favour of the two-stage approach (Romano et al., Knee Surg Sports Traumatol Arthrosc 2012;20(12):2445-53). All three of these reviews have important methodological limitations and/or highlight the low quality of the pooled evidences available to them at the time of their synthesis exercise. The lack of agreement between them prior to our updated review supports the concept of equipoise between the interventions.

4. Methods/p.7: line38-40. As mentioned in the limitations, there is no definition of PJI in this study (see p.15).

RESPONSE:

Although consensus opinions exist on the criteria which should be used to diagnose a PJI, there is no gold standard test or universally accepted diagnostic criteria to determine the presence or absence of PJI. The definition of PJI is at the discretion of each surgeon generally following local procedures adapted from the AAOS recommendations. Given that a patient undergoes revision surgery for PJI according to the opinion of the treating surgeon, rather than due to having a positive test or meeting a certain set of criteria, when considering the changing rates, trends and surgical burden of revision for PJI we feel that the surgeon defined diagnosis of PJI is of paramount importance. Our work is therefore based on a pragmatic definition of PJI using the indication reported by surgeon at the time of surgery, a reflection of their "intention-to-treat" approach. We have now extended the methods section to reflect this:

"Information on microbiology results are not recorded in the NJR and infected index procedures were identified using subsequent revision performed for an indication of PJI. The diagnosis and treatment strategy for PJI is at the discretion of the surgeon according to local protocols and available information at the time of surgery."

5. Methods/p.7: line 40. The definition of "revision procedures" is not complete, since debridement with/without exchange of the modular parts of the device is also a revision procedure. The number of patients undergoing this procedure should also be given. In addition, the number of patients with PJI not undergoing any surgery at all, but only palliative antibiotic suppression should be given, otherwise the prevalence of PJI is underestimated.

RESPONSE:

We agree that any further operation a patient undergoes represents a surgical intervention. The NJR does not capture surgeries where there is no removal, addition or change of an implant or accessory and therefore debridements without modular exchange are not captured and are beyond the scope of this work. As previously mentioned we have now reported DAIRs with modular exchanges captured in the NJR through analysis of the component level data and these are reported in table 2. Likewise, non-surgical treatment of PJI is not recorded in the NJR. For this reason our work does not report on the prevalence or incidence of PJI but on the rates of revision for PJI, especially revision for PJI requiring total or partial implant exchange. The methods section (Index surgeries and revision surgeries for PJI paragraph) has been modified to further clarify the procedures captured in the NJR.

"Information on microbiology results are not recorded in the NJR and infected index procedures were identified using subsequent revision performed for an indication of PJI.

...

Non-surgical management of PJI and surgical procedures where no revision or modular exchange of implants is performed are not recorded in the NJR.

Revision procedures are reported in the NJR as a single stage, a stage one of a two-stage revision, a stage two of a two-stage revision procedure, a conversion to arthrodesis, or an amputation. DAIR procedures with modular exchange are recorded in the NJR dataset as single stage revision procedures. To identify which procedures recorded as single stage revisions were DAIRs with modular exchange as opposed to complete single stage revisions where implants fixed to bone are also revised, the component level data was considered for the index and revision procedures. Implant component labels are compulsory for all NJR records, the minimum dataset forms also contain fields for components removed but this information is discretionary and limited to the brand removed, therefore does not allow component tracing unless a linked index procedure exists. Procedures were therefore defined on the basis of the data provided for components implanted at revision surgery.

Procedures recorded as single stage revisions where only modular components were added (termed "meniscal component" in the minimum data set) were defined as DAIRs with modular exchange.

Those where implants fixed to bone were implanted ("femoral component", "tibial tray" +/- "patella")

were defined as single stage revisions. Debridements where the surgeon either elects to not exchange modular components when they are present or where modular exchange is not possible (e.g. with monoblock polyethylene tibial components) are not captured in the NJR.”

6. Results/p.12: lines 3-5. The observation “Revision rates within 3 months of index procedure increased over time...” is very interesting. In the Discussion, possible reasons for this apparent increase should be analyzed (more rapid diagnosis? Better contact with rehabilitation centers leading to more rapid readmission? Less suppressive antibiotics without revision?).

RESPONSE:

We have slightly extended our discussion but have been cautious to avoid speculating beyond what our data and findings support and would allow us to reasonably comment on:

“This phenomenon is likely to be multifactorial. Factors that could lead to an increased diagnosis of PJI in this period and hence increased risk of revision include the increased accuracy of tests available to clinicians for the diagnosis of PJI,[33] coupled with more rapid diagnosis and/or treatment in specialist centres, the increased risk factors for PJI amongst the population undergoing knee replacement population (such as elevated body mass index [34]) and increased bed occupancy within the healthcare setting in which these procedures were performed.[35] There may be a trend towards the use of revision surgery to manage PJI rather than suppressive treatment with antibiotics but we cannot comment on this as non-surgical management of PJI is beyond the scope of this study. There have been similar findings in other countries that could not be accounted for by risk factors recorded in those registries, suggesting this trend could reflect an actual general increase in the risk of PJI. [36] It is interesting to note from our results that there has not been a relative increase in the use of DAIRs with modular exchanges in the NJR to explain this phenomenon.”

7. Discussion/p.14: line 54. The statement “...utilisation of single stage revision could be safely increased” is somewhat dangerous, since the type of procedure should be carefully chosen according to an algorithm based on risk factors for the specific procedure. E.g. patients with sinus tracts or difficult-to-treat microorganisms should not be treated with 1-stage exchange.

RESPONSE:

As discussed earlier, up to date meta-analysis of the literature has shown no significant difference in rates of failure due to reinfection for unselected cases in single vs. two-stage revision surgery for the treatment of PJI of the knee (Kunutsor et al., PloS One 2016;11(3): e0151537). Whilst we acknowledge the work of Professor Zimmerli in the development and successful application of the algorithm he mentions, the most comprehensive and therefore generalisable review of the literature to date demonstrates no difference in reinfection rates between the treatment strategies and hence justifies the statement that utilisation of single stage revision could be safely increased.

The sentence “...utilisation of single stage revision could be safely increased” has now been removed from the first section of the discussion and the last section of the discussion (Unanswered questions and future research) has been extended as follow: “Although the utilisation of single stage revision knee replacement for the treatment of PJI has increased to approximately 20%, given the current evidence base shows equivalent reinfection rates following single stage compared to two-stage revision surgery for PJI, [19 48] it might be possible that the proportion of patients treated with single-stage surgery taking account of patient, surgeon, and operative factors could be further increased. Given the increasing burden of treating PJI, this may help ameliorate the increase in resources that will otherwise be required to treat this condition [49]. There is currently no data available from randomised controlled trials looking at the outcome of these two treatment strategies and such evidence is required to support the decision making process in the planning and treatment of these cases.”

8. Discussion/p.15: lines 37-46. The statement that “changing only modular components is not distinct from single stage surgery...” is not correct. In case of single stage exchange, the surgeon tries to

eliminate the whole biofilm, which cannot be done when only modular parts are exchanged.

RESPONSE:

This comment refers to the recording of procedures in the NJR where DAIRs with modular exchange are currently recorded as single stage procedures. As we have described above, we have separated out DAIRs with modular exchanges from true single stage revision by analysis of the component level data associated with individual cases. The sentence has been changed as follow:

“Currently in the NJR, the only option to record DAIR procedures where modular exchange is performed is to record them as single stage revision surgeries.”

9. Discussion/p.17: lines 24ff. As mentioned above (#6), the discussion of the “increase...in the first 3 months...” is not complete at all.

RESPONSE:

Please see our response to point 6.

Reviewer: 3

Reviewer Name: Ove Furnes

Institution and Country: Department of orthopaedic surgery, Haukeland University Hospital, Norway

Please state any competing interests: Head the Norwegian Arthroplasty Register

We would like to thank Professor Furnes for his comments and suggestions.

It is an important study from the largest joint registry in the world. I have some suggestions and questions to the authors.

The paper is difficult to read because of the complexity of the issue with prosthetic joint infections (PJI) and the terminology used.

1. Figure 1 The term: recorded index surgery is non-intuitive. To me index implies a primary operation but here it means both primary and subsequent revisions as long as they are not done due to infection. This should be explained. In the last row : performed between 01/2004-31/2014 the numbers of primary and aseptic revisions should be given.

RESPONSE:

We do apologise for this confusion. We have now modified figure 1 to better reflect our sample and define the term “index surgery”. This term refers to any procedure at risk of being revised for PJI whether this is a primary or a revision procedure, as long as the prosthesis of interest has never been revised for an infection.

Very few papers have investigated revision for PJI following index “aseptic” revision and this manuscript aims to provide descriptive epidemiological findings to fill this gap.

2. In the abstract under the design heading 8031 revision total knee replacements are used, in table 2 the number of 8658 are used, in figure 1 I count 8302 PJI adding total revision for PJI. Explain this discrepancy.

RESPONSE:

Again, we apologise for this confusion. We have amended the abstract to reflect the figures provided in figure 1 and provided further details in figure 1 to help the comprehension of our working samples:

“The cohort analysed consisted of 679,010 index primary knee replacements, 33,920 index revision knee replacements and 8,247 revision total knee replacements performed due to a diagnosis of PJI.”

We have also modified the discussion for consistency:

“It includes 712,930 primary and aseptic revision index procedures and 8,247 revision total knee replacements performed due to a diagnosis of PJI.”

For the sake of clarity 8,031 has been replaced by 8,247 in the abstract. These three sets of figures (8,031, 8,658 and 8,302) correspond to the previous version of the manuscript. They have changed as the analysis of component level data to identify the DAIR procedures with modular exchange has allowed us to reclassify revision procedures inappropriately classified as single-stage into stage one of a two-stage revision. The new totals are 7,976, 8,587 and 8,247 respectively, as the number of single-stage has decreased and some of the reclassified procedures have completed two-stage procedures for which only the stage two was reported (stage one and stage two of a two stage procedure are only accounted as one procedure)

The old and new totals stand as:

-8,031: $4,026+789+3,216$ (figure 1), the sums of revisions for PJI performed on patients with an index procedure done between 2003-2013+the unlinked procedures for PJI described in section B of figure 1. The new total is $7,976:4,004+785+3,187$.

-8,302: $8,031+100+22+149$. This number also includes the revisions for PJI performed in 2014 unlinked to an index procedure or linked to an index procedure performed in 2014. The new total is $8,247: 7,976+100+22+149$

- 8,658: $8,302+163+43+150$. This is the overall burden of PJI reported in the NJR between 2003-2014. It includes the 356 aseptic re-revisions performed on patients already "revised" for PJI. This number has been clarified in figure 1. The new total is $8,587: 8,247+153+42+145$

3. The use of the term prevalence and cumulative revision risk is confusing for me. The authors use them differently in the results in the abstract, tables, figures and in the text in the results section. In the result section of the abstract the revision risk was 0.5% following primary and 2.1% following revision knee replacement. Do the authors mean prevalence or cumulative revision risk? Is the revision risk calculated after 1, 5 or 10 years following primary and revisions procedure? In the result section the authors refer to Table 1 for the revision rate but in this table the prevalence rate are given (5.39 per 1000). Is this the same as the revision risk of 0.5% in the abstract? I had expected that you used numbers from figure 4 as risk for revision. I would like to read the tables and figures and see the same numbers in the text. This is a general issue with the paper.

RESPONSE:

Thank you for this comment. Reporting the prevalence, incidence and the total burden of revision for PJI in the same manuscript is challenging but important as each brings relevant descriptive facts, respectively for surgeons, patients and the health care system. The figures presented in the text were rounded. We have reformatted the reporting of the figures in the results and abstract sections to be clearer and coherent with the tables and figures content.

For example:

"Of the primary knee replacements recorded in the NJR, 5.39/1,000 (95%CI 5.21-5.56) were subsequently revised due to PJI (Table 1). Of the aseptic revision knee replacements, 21.14/1,000(95%CI 19.61-22.67) were subsequently revised due to PJI."

4. Another issue is the lack of distinction between DAIR (surgical debridement, antibiotic and implant retention), one stage and two stage revision for PJI. DAIR is discussed on page 15. It should be mentioned more clearly in the methods section. Some papers find up to 70% success with DAIR. Patients with acute PJI would usually go through a debridement with exchange of easily exchangeable parts like tibia polyethylene inserts. How do you handle an AGC monoblock knee if it is revised due to infection with DAIR. Will this procedure be included in the register compared to a NexGen modular fixed bearing knee. How do you distinguish this procedure from a one stage revision? Can NJR distinguish between removal of implant, insert of implant, exchange of implant or implant parts, insert of cement spacer, exchange of polyethylene insert? This should be explained in the methods section (See Engesæter LB. Surgical procedures in the treatment of 784 infected THAs reported to the Norwegian Arthroplasty Register. *Acta Orthop* 2011;82 (5):530-537.

RESPONSE:

In the NJR, DAIR procedures with retention of the implant and no addition, removal or change of implants or accessories are not captured. The only option to record a DAIR procedure with modular exchange in the NJR is as a single stage procedure. We have now analysed the individual case component level data to separate out DAIRs with modular exchange from the true single stage procedures where femoral and tibial implants fixed to bone have been changed; these DAIRs with modular exchange are identified and reported in table 2. Implant component labels are compulsory for all NJR records, the minimum dataset forms also contains fields for components removed but this information is discretionary and limited to the brand removed, therefore does not allow component tracing unless a linked index procedure exists. Procedures were therefore defined on the basis of the component data provided for components implanted at revision surgery. Procedures recorded as single stage revisions where only modular components were added (termed "meniscal component" in the minimum data set) were defined as DAIRs with modular exchange. Those where implants fixed to bone were implanted ("femoral component", "tibial tray" +/- "patella") were defined as single stage revisions. Debridements where the surgeon either elects to not exchange modular components when they are present or where modular exchange is not possible (e.g. with monoblock polyethylene tibial components) are not captured in the NJR.

As a result of this new investigation we have identified 110 revisions for PJI which were previously classified as single-stage according to the data submitted by the treating surgeon (customised-made cement spacer with or without various accessories). They have now been converted into a stage one of a two-stage procedure which has slightly modified the numbers reported in table 2 and appendix table 1 as the two stages of a two-stage procedure are only accounted once in table 2.

We have modified the methods, results and discussion section to reflect our strategy, present those new findings and discuss the associated limitations.

In the methods section:

"DAIR procedures with modular exchange are recorded in the NJR dataset as single stage revision procedures. To identify which procedures recorded as single stage revisions were DAIRs with modular exchange as opposed to complete single stage revisions where implants fixed to bone are also revised, the component level data was considered for the index and revision procedures. Implant component labels are compulsory for all NJR records, the minimum dataset forms also contain fields for components removed but this information is discretionary and limited to the brand removed, therefore does not allow component tracing unless a linked index procedure exists. Procedures were therefore defined on the basis of the data provided for components implanted at revision surgery. Procedures recorded as single stage revisions where only modular components were added (termed "meniscal component" in the minimum data set) were defined as DAIRs with modular exchange. Those where implants fixed to bone were implanted ("femoral component", "tibial tray" +/- "patella") were defined as single stage revisions. Debridements where the surgeon either elects to not exchange modular components when they are present or where modular exchange is not possible (e.g. with monoblock polyethylene tibial components) are not captured in the NJR."

In the results section

"The DAIR procedures represented around 5% of the total reported procedures and 9% of those reported in 2014."

In the discussion:

"Currently in the NJR, the only option to record DAIR procedures where modular exchange is performed is to record them as single stage revision surgeries. [25] This complicates their differentiation from single stage surgeries where the femoral and tibial components are revised. This represents a potential weakness of the data collection form as this may be subject to different interpretation by surgeons, despite the fact that recording of procedures in which any component is removed or inserted is mandatory. We have used component level data for individual cases to identify

DAIR procedures with modular exchange. A few single-stage revisions for PJI unlinked to an index procedure were considered as DAIRs in view of the components implanted (n=66).”

5. Datasource Paragraph 2: 1 st april or 1.january??

Index procedures should be explained including both primary and aseptic revisions. The explanation come after the term is introduced.

RESPONSE:

Thank you for this comment. The typographical error has been corrected. The method section has been reformatted to allow a better presentation of the index and revision procedures.

“Index surgeries and revision surgeries for PJI

We grouped procedures as index knee replacements and revision surgeries for treatment of knee PJI. Index surgeries included all primary procedures and all revision procedures performed for an indication other than infection. The index revision procedures have been labelled “aseptic revision” procedures to indicate they were not performed due to PJI. All index surgeries performed between 1st April 2003 and 31st December 2013 were included (to allow a minimum 12-months follow-up). Information on microbiology results are not recorded in the NJR and infected index procedures were identified using subsequent revision performed for an indication of PJI. The diagnosis and treatment strategy for PJI is at the discretion of the surgeon according to local protocols and available information at the time of surgery. Revision surgeries performed as a consequence of PJI between 1st April 2003 and 31st December 2014 were considered. Revisions not performed for PJI but that were performed on a knee previously operated on due to PJI were not considered as index procedures, but were used alongside the revision and re-revision procedures for PJI to define the “burden of PJI”. Non-surgical management of PJI and surgical procedures where no revision or modular exchange of implants is performed are not recorded in the NJR.”

6. Statistical analyses. It is stated that 85% and 99% of knee replacements were captured. Indicate if these are primary cases or all, and give an estimate for the capture percentage of revisions.

RESPONSE:

These figures are the proportion of procedure records submitted to the NJR compared with the levy returns for the number of implants sold. Specific figures by type of procedures are unavailable to us and the conduct of such investigations is beyond the scope of this work.

The text has been modified as follow:

“This period was selected as over 85% (proportion of procedure records submitted to the NJR compared with the levy returns for the number of implants sold) of knee replacements performed in 2005 and over 99% of those performed from 2007 onwards had been recorded in the NJR; prior to 2005, the data capture of the NJR was <75%.[21]”

7. Under patient perspective the authors state that competing risk of death and revision for an aseptic indication was accounted for. In the Fine and Grey model one can usually account for only one competing risk cause at the time. How did you manage this?

RESPONSE:

The cumulative incidence functions (CIF) are not calculated with the Fine and Grey model. We have used the “stcompet” command proposed by Stata and the reader will be able to find more practical and theoretical information in the reference Coviello et al. listed in our bibliography. More practical details are also provided in the help section of this command with a list of useful references detailing the theory behind this command (in particular Choudhury et al.)

8. Results:

Figure 2 show increasing prevalence of PJI over time but table 1 show a decrease in prevalence by time. This discrepancy needs explanation?

RESPONSE:

A prevalence rate is a useful statistic for surgeons as it describes the proportion of procedures they have performed which have been revised for an infection. However, yearly prevalence rates have a limited interest per se. A year-specific prevalence rate does not account for the length of time an index procedure has been followed up. Earlier procedures have been "at risk of being infected" for a longer period of time and hence the higher number of procedures revised for PJI and higher prevalence rates compare to more recent period with shorter follow-up. Table 1 serves two purposes. First to describe our sample by year and type of procedures, specifically the number of procedures and those revised for PJI. The second purpose is to present the overall prevalence rate for the entire period of observation, to provide the surgeon perspective. Annual prevalence rates have more sense when presented as per figures 2, 3 (and their related appendix figures) as we have identified the prevalence rate of procedures revised for a specific period from the index surgery and limited the presentation of the results for the specific year for which the studied "period of revision" could have been observed, making annual rates directly comparable and allowing us to identify time trends. The relevance of this approach has been acknowledged and further detailed by reviewer 4. Prevalence rates are useful for surgeons but should be supplemented with incidence rates to provide a more patient-centred perspective. This is why we have presented the cumulative incidence function (~incidence rate) to account for the 1. length of time each patient had been at risk of being revised for an infection, and 2. competing risks .

9. Patient perspective

Probability of revision at 5 years was 0.56% and 2.41% which is not the same numbers as in the abstract. Explain.

RESPONSE:

We apologise for this disparity and have now amended the abstract to reflect the incidence rates/probability derived from the CIFs rather than rounded prevalence rates.

"The incidence of revision total knee replacement due to PJI at two years was 3.2/1,000 following primary and 14.4/1,000 following revision knee replacement respectively."

10. Table 2 does not attempt to distinguish between one stage and DAIR where component parts are exchanged.

RESPONSE:

We have now modified table 2 to report on those procedures. Please see explanations provided in point 4.

11.a Discussion:

In the discussion under limitations and strengths it should be mentioned if NJR have validated the revision procedures and in particular PJIs

(See Gundtoft PH et al. Validation of the diagnosis prosthetic joint infection in the Danish Hip Arthroplasty Register. *BJJ*;98-B:320-5.

Gundtoft PH et al. One-year incidence of prosthetic joint infection in total hip arthroplasty: a cohort study with linkage of the Danish Hip Arthroplasty Register and Danish Microbiology Database. *Osteoarthritis and Cartilage* 2016 epub.

Lindgren JV et al. Validation of reoperations due to infection in the Swedish Hip Arthroplasty Register. *BMC Musculoskeletal disorders* 2014 Nov 19;15:384).

RESPONSE:

Thank you for this recommendation which highlights the need for further work and consideration of the management of PJI in arthroplasty. This is likely to also be a limitation of the data available from the NRJ and we have now stressed it in the discussion with the references mentioned above. We do

consider our estimations of the burden of PJI as conservative figures and the cost for the NHS is likely to be even higher. We have added the following sentence to the discussion:

“The capture of revision surgeries, in particular those performed for PJI is also not perfect in arthroplasty registries [26-29] and it is unclear how complete the capture of these procedures in the NJR is.

The annual burden of PJI with over 1,000 procedures in recent years, while already high and expensive, is a conservative estimation and the cost of PJI considerable for the NHS.”

11.b Discussion: According to the NJR reports the yearly revision rate has increased since 2003. I have interpreted this as an increased completeness of reporting to NJR for revision operations. Could this be commented.

RESPONSE:

The completeness of the revision procedures in the NJR seems to have increased over time. This could be a hypothesis to explain in part the increase in the early revision for infection, but it would be surprising if this were the only or main explanation. There is no reason to consider that the increase in the reporting of revisions would be limited to revision procedures performed in the early period after an index operation. It is more likely that the increase of reporting would have affected all revisions, at least for those in the first 2 to 5 years following the index surgery. In this case, this would have affected most of the periods of revision for infection, certainly the revisions within the first 2 years. Our findings do not highlight any trend beyond 3months post-index surgery.

12. Can the increased use of modular implants be a cause of the increased revision rate recorded for PJI in NJR?

RESPONSE:

Only a very small proportion of total knee replacements recorded in the NJR dataset are non-modular designs. These represent 1.3% of the recorded primary knee procedures in the dataset since April 2003. As has been restated in the methods section in response to previous comments, debridements that do not involve the change of any components are not captured in the NJR. Whilst this is a limitation created by the nature of the data capture in the NJR, we believe the overall effect of this is likely to be small.

Reviewer: 4

Reviewer Name: Gregory J. Stoddard

Institution and Country: University of Utah School of Medicine, USA

The authors have done a nice job of demonstrating the revision issue is a 0 to 3 months post-surgery phenomenon. In their Table 1, the prevalence of revisions is decreasing across time, suggesting, at first, that we are doing surgeries better as time goes on. In Figures 2 and 3, we see a relationship between revisions and temporal time in four panels, where the panels represent the time between surgery and revision. It is noticeable that the longer the time between surgery and revision, the flatter the trend line across years. So, revision risk is a 0 to 3 months post-surgery issue, and we are not getting better at reducing that risk. The prevalence of revisions across time in Table 1 is going down, but that is because >3 months data are being added to the denominator faster than the 0 to 3 month data. The authors recognized all of this, and the way they reported their conclusions correctly reflects it. I have no revision suggestions. Nice work!

RESPONSE:

We would like to thank Dr Stoddard for his very positive comments. They are providing relevant elements to answer point 8 of reviewer 3

VERSION 2 – REVIEW

REVIEWER	Hemant Pandit University of Leeds, UK
REVIEW RETURNED	24-Feb-2017

GENERAL COMMENTS	Very nicely written paper with a clear message. I have no specific comments to add.
---

REVIEWER	Werner Zimmerli, MD, Professor Consultant Interdisciplinary Unit for Orthopaedic Infections, Kantonsspital Baselland, Liestal, Switzerland
REVIEW RETURNED	06-Mar-2017

GENERAL COMMENTS	In this revised manuscript, the authors have very carefully considered the comments of all reviewers. The modifications are as good as they can be with the available data from the National Joint Registry. The separation of cases with DAIR from those with one-stage exchange allows now a better comparison with similar studies in which all types of surgical treatments are reported. I have only three remaining comments regarding the revised manuscript. Specific comments. 1. P 61/84, i.e. P.5/L.40 of R1: The statement “For patients with PJI after knee replacement who are diagnosed before biofilm formation occurs...” is not correct, since biofilm formation starts immediately after contact of bacteria with implant material. However, young biofilms (up to about 3-4 weeks) are obviously still susceptible to biofilm-active antibiotics such as rifampin against Gram-positive cocci or fluoroquinolones against Gram-negative bacilli. Thus, the following statement would be more appropriate: “For patients with acute PJI (first month after implantation or <3 weeks after hematogeneous seeding), DAIR with modular” 2. Comment 7 to reviewer 2 and P.62/84: In most studies, the statement “unselected cases” is not correct regarding the decision to treat with one-stage exchange. E.g. in the paper of Tibrewal S et al. (BJJ 2014), “intact soft-tissue cover of the knee” was required. I didn’t check, but I am convinced, that this requirement for one-stage exchange was considered in all studies. Thus, a randomized controlled trial comparing one-stage vs two-stage exchange would ethically be problematic. This fact should be added in the first paragraph on P.62/84. 3. P.69/84, L.7. I guess that the statement “...RR 246 (95%CI 1.15-5.25...)” is a typing error and should be read as “...RR 2.46...”
---

REVIEWER	Ove Furnes Haukeland Universityhospital Bergen, Norway
	None declared Head The Norwegian Arthroplasty Register
REVIEW RETURNED	28-Feb-2017

GENERAL COMMENTS	I am happy with the review and changes in the manuscript
--

VERSION 2 – AUTHOR RESPONSE

We would like to thank the reviewers for their comments. Professor Zimmerli's comments have been addressed below.

1. We have modified the sentence as follow

"For patients with acute PJI diagnosed before biofilm maturation occurs, DAIR with modular exchange is a reasonable treatment option.... "

2. We have removed the reference to the absence of data from randomised trials from the background section, and noted that the evidences from cases-series are from a pooled analysis. Moreover, we have modified the last section of the discussion to highlight the potential value of comparisons of surgical method within registry.

Introduction

"Pooled evidence from case-series suggests similar rates of re-infection after each treatment. The decision to treat with a single-stage or two-stage procedure may be guided by microbiological tests, patient and surgical factors but ultimately the choice of revision method is largely that of the treating surgeon."

Discussion

"In the absence of data from randomised controlled trials or other direct comparisons, we plan further analyses of the NJR data which will compare outcomes after one- and two-stage methods with robust adjustment for key patient and surgical factors. Such evidence will support the decision making process in the planning and treatment of patients with PJI after knee replacement."

3. The typing error has been modified: RR 2.46 rather than RR 246.

VERSION 3 – REVIEW

REVIEWER	Werner Zimmerli, MD, Professor Interdisciplinary Unit for Orthopaedic Infections, Kantonsspital Baselland, Liestal, Switzerland
REVIEW RETURNED	11-Mar-2017

GENERAL COMMENTS	The authors adequately considered the 3 criticisms in the review of the R1-version. I have no further suggestions.
--